

# Linking patient-reported oral and general health-related quality of life

Danna R. Paulson[1], Phonsuda Chanthavisouk[1], Mike T. John[2,3],
Leah Feuerstahler[4], Xing Chen[4] and Aparna Ingleshwar[2]

[1] Department of Primary Dental Care, School of Dentistry, University of Minnesota, Minneapolis, MN, United States
[2] Department of Diagnostic and Biological Sciences, School of Dentistry, University of Minnesota, Minneapolis, MN, United States
[3] Division of Epidemiology & Community Health, School of Public Health, University of Minnesota, Minneapolis, MN, United States
[4] Department of Psychology, Fordham University, Bronx, NY, United States

Corresponding author
Danna R. Paulson,
dannardh@umn.edu

## ABSTRACT

**Background:** The relationship between oral and overall health is of interest to health care professionals and patients alike. This study investigated the correlation between oral health-related quality of life (OHRQoL) and health-related quality of life (HRQoL) in a general adult population.

**Methods:** This cross-sectional study used a convenience sample of adult participants (N = 607) attending the 2022 Minnesota County and State fairs in USA, the 5-item Oral Health Impact Profile (OHIP-5) assessed OHRQoL, and the 10-item PROMIS v.1.2 Global Health Instrument assessed HRQoL. Spearman and Pearson correlations were used to summarize the bivariable relationship between OHRQoL and HRQoL (both physical and mental health dimensions). A structural equation model determined OHRQoL-HRQoL correlations (r). Correlations' magnitude was interpreted according to Cohen's guidelines (r = 0.10, 0.30, and 0.50 to demarcate "small," "medium," and "large" effects, respectively).

**Results:** OHRQoL and HRQoL correlated with r = 0.52 (95% confidence interval, CI: [0.50–0.55]), indicating that the two constructs shared 27% of their information. According to Cohen, this was a "large" effect. OHRQoL, and the physical and mental HRQoL dimensions correlated with r = 0.55 (95% CI: [0.50–0.59]) and r = 0.43 (95% CI: [0.40–0.46]), respectively, indicating a "large" and a "medium" effect. OHRQoL and HRQoL were substantially correlated in an adult population.

**Conclusion:** Using OHIP-5 to assess their dental patients' oral health impact allows dental professionals to gain insights into patients' overall health-related wellbeing.

## INTRODUCTION

Oral health plays a pivotal role in daily life, influencing various aspects of well-being. From basic functions like eating and speaking to social interactions and self-esteem, the condition of one's oral health significantly impacts life. Tooth pain can impair the ability to

chew food properly, leading to dietary limitations and nutritional deficiencies. Similarly, oral conditions such as gum disease or missing teeth can affect speech, appearance, and confidence. The appearance of one's face and smile can greatly influence self-perception and interpersonal relationships. Beyond these immediate effects, untreated oral disease can contribute to systemic health problems, including cardiovascular diseases and diabetes, further underscoring the importance of maintaining good oral health for overall well-being (*Nazir, 2017*; *Dörfer et al., 2020*). Oral health is interconnected to various facets of daily life, underscoring the significance of understanding this link.

Examining the intricate relationship between oral health-related quality of life (OHRQoL) and health-related quality of life (HRQoL) is vital for complete health assessments. While the influence of poor oral health on HRQoL and vice versa is acknowledged, the magnitude of this connection remains unknown. This gap is particularly significant as it represents a comprehensive summary of the broader health-oral health relationship.

Oral diseases impact patients in one or more of the four OHRQoL dimensions; Oral Function, Orofacial Pain, Orofacial Appearance, and Psychosocial Impact which represent the elemental building blocks of OHRQoL (*John et al., 2014*). The same applies for diseases which impact HRQoL in its two dimensions, Physical Health and Mental Health. Therefore, the magnitude of the OHRQoL-HRQoL correlation could be different depending on the oral and general diseases being experienced by the patient, and the population being studied.

Studies focused on general adult populations consistently highlight a positive and statistically significant correlation between OHRQoL and HRQoL, although there are considerable variations (*Zimmer et al., 2010*; *Reissmann et al., 2013*; *Sekulić et al., 2020*). For example, German dental patients and the German general population demonstrated correlations between OHRQoL and HRQoL of r = 0.24 and r = 0.28 (per Cohen's correlation benchmarks, this indicates a moderate correlation), respectively, using the 49-item Oral Health Impact Profile (OHIP-49) and Short-Form 23 (SF-35) with a 1-month recall period and structural equation models (SEMs) (*Cohen, 1988*, *1992*; *Reissmann et al., 2013*; *Naik et al., 2016*). Conversely, among adult dental patients of a nonprofit health care provider in Minnesota (USA), OHIP-49 and Patient-Reported Outcomes Measurement Information System (PROMIS) v.1.1 Global Health Instrument using a 1-month recall period produced a SEM-derived correlation of r = 0.56 (*Sekulić et al., 2020*). Further dividing the OHRQoL-HRQoL relationship into a mental health-OHRQoL component and a physical health-OHRQoL component, and using bivariable correlations with Pearson correlation coefficients, identified correlations of r = 0.45/0.45 (*Reissmann et al., 2013*), r = 0.47/0.52 (*Sekulić et al., 2020*), and r = 0.31/0.32 (*Zimmer et al., 2010*) among a German general population, American dental patients, and German dental patients, respectively (*Zimmer et al., 2010*; *Reissmann et al., 2013*; *Sekulić et al., 2020*).

Disease-specific research in this domain has also consistently emphasized the substantial correlation between OHRQoL and HRQoL. Studies involving temporomandibular disorder (TMD) patients, head and neck cancer patients, and

individuals with dentofacial deformities all demonstrate positive correlations between OHRQoL and HRQoL (*Öhrn et al., 2001*; *Balik, Peker & Ozdemir-Karatas, 2021*; *Duarte et al., 2022*). Further substantiating this connection, another study among community-dwelling elders unveiled negative correlations between OHIP-14 scores assessing OHRQoL and all four domains of the World Health Organization Quality of Life Assessment (WHOQoL-Bref). Statistical significance emerged particularly within the physical and mental HRQoL domains (*Kuo et al., 2018*). Collectively, these studies emphasize the considerable magnitude of the correlation between OHRQoL and HRQoL, underscoring the pivotal intersection of oral health and overall well-being.

Existing studies use different instruments to measure the constructs which include the OHIP-5, -14 and -49 for OHRQoL measurement and the Short Form -12, and -36, as well as the PROMIS General Health instrument v.1.1 and v.1.2 to measure HRQoL. These variations are influenced by the measurement instruments used for OHRQoL and HRQoL, including recall period lengths and instrument length. For example, several OHRQoL instruments exist, and even with respect to the most widely used instrument, the OHIP, there is considerable variation in terms of the length of the recall period, and the length of the instrument, *i.e.*, the number of items included, across its different versions (*Slade & Spencer, 1994*; *Waller et al., 2016*; *John et al., 2022*; *Ingleshwar & John, 2023*). Different analytical approaches have also been taken, as some studies report correlations of the two constructs, whereas others report SEM models. Additionally, the populations differ; from community settings to general dental, and even oral-disease specific populations. Because the concepts OHRQoL and HRQoL are tied to a time period when individuals experience the oral health impact, instrument recall periods affect the correlation. Longer recall periods are especially interesting for characterization of population health compared to shorter recall periods which are more relevant for clinical applications. Studies using a 12-month recall period are not available. Analytical approaches also shape correlation magnitudes, with bivariable correlations using instrument summary scores, and SEMs taking the latent variable nature of HRQoL and OHRQoL into account. Finally, while chance, *i.e.*, sampling variability, adds to the complexity of correlation findings, the OHRQoL-HRQoL correlation can be truly different in varying populations.

The relationship between OHRQoL and HRQoL forms a critical foundation for comprehensive healthcare evaluations. The existing body of research, while acknowledging the shared influence of oral health and overall well-being, has demonstrated variations in the magnitude of this connection across diverse populations and measurement methodologies. This study aimed to determine the OHRQoL-HRQoL relationship over a longer period, employing a 12-month recall period, the OHIP-5, and the 10-item PROMIS 1.2—Global Health Instrument within an adult community sample to provide a deeper understanding of the OHRQoL-HRQoL relationship within an adult community sample. Using an extended recall period, we endeavor to fill current gaps in understanding, providing updated evidence on the OHRQoL-HRQoL relationship and informing modern healthcare practices. Despite the advantages of the practical, valid, and reliable 5-item version of the OHIP, to our knowledge, there is a gap using this practical instrument to

assess the relationship between OHRQoL and HRQoL over a longer period. We aim to provide valuable insights and raise awareness among healthcare professionals regarding the significant impact of oral health on individuals' overall well-being. By emphasizing the interconnectedness of oral health with general health, we strive to facilitate the development of more comprehensive healthcare strategies.

## MATERIALS AND METHODS

### Participants and study design

During the months of July and August of 2022, participants were conveniently sampled for participation in a cross-sectional study. Recruitment took place in-person at two prominent Minnesota county fairs and at the Minnesota State Fair which is the second largest state fair in USA. Notably, the state fair garnered an impressive attendance of 1,842,222 individuals over its 12-day duration in 2022, averaging more than 150,000 attendees per day (*Minnesota State Fair, 2022*). Interested fairgoers voluntarily approached the study booth to express their willingness to participate. Adults aged 18 years and older who were proficient in English and capable of participating in a brief oral screening involving opening and holding the mouth open for approximately 2 minutes were included. Exclusion criteria comprised of those lacking capacity to consent. Following a comprehensive explanation of the study and determination of inclusion/exclusion criteria, participants underwent the informed consent process, written consent, enrollment, and subsequently completed the study activities. Ethical oversight was ensured by the University of Minnesota (UMN) Institutional Review Board (IRB), which granted approval for all study procedures under the study ID: STUDY00016028. Both OHRQoL and HRQoL instruments were self-administered by the participants, with the research team's assistance, as necessary. For instance, the research team read the survey items out loud to a visually impaired participant, and recorded responses on the participants' behalf. Survey responses were collected electronically on tablets using the research electronic data capture system (REDCap) (*Harris et al., 2009*).

### Physical oral health measurement

The physical assessment included various measurements, starting with the quantification of physical oral health through a simple count of a participant's natural teeth by a calibrated study team member who was either a dentist, dental hygienist, dental therapist, or allied oral health student at the UMN School of Dentistry. The team members responsible for conducting the teeth count underwent comprehensive training using a study training guide, which provided detailed instructions on the indicators for inclusion as a "present" tooth (such as third molars, retained root tips and primary teeth) and indicators for exclusion from the tooth count (such as implants, pontics, and implant-supported dentures). This training ensured consistency and accuracy in the physical oral health measurements across all participants.

## Oral health-related quality of life (OHRQoL) measurement

The Oral Health Impact Profile (OHIP) is a valid and reliable instrument that has been thoroughly psychometrically tested. It has been shown to accurately measure the construct of OHRQoL, and its four dimensions (*John et al., 2014*). The OHIP was originally developed as a 49-item instrument, the OHIP-49 (*Slade & Spencer, 1994*). Over time, researchers have created numerous short-forms and translations of the instrument. Regarding practicality, a five-item instrument is more feasible for use in all settings when compared to the 49-item version. Although longer instruments tend to measure OHRQoL better because of improved reliability, the 5-item version of the Oral Health Impact Profile (OHIP-5) instrument has been shown to reliably capture 86% of all OHRQoL variability captured by the original 49-item version, the OHIP-49 (*Naik et al., 2016*; *John, 2022*; *John et al., 2022*). For this reason, the four dimensions of OHRQoL were measured through the use of the concise OHIP-5. Response options for each of the five items include 0 (never), 1 (hardly ever), 2 (occasionally), 3 (fairly often), and 4 (very often). OHIP-5 summary scores can range from zero to twenty. Traditionally, an OHIP-5 summary score of "0" would indicate that the participant has not had any perceived oral health impacts, or problems. Higher scores indicate more self-reported oral health impacts, resulting in a decreased OHRQoL. For the purpose of this study, OHIP-5 scores were reverse coded so that greater summary scores signify better OHRQoL to align with the coding of HRQoL.

## Health-related quality of life (HRQoL) measurement

The Patient-Reported Outcomes Measurement Information System (PROMIS) version 1.2—Global Health instrument is a 10-item, psychometrically sound, and validated instrument developed using Item Response Theory (IRT) (*Patient-Reported Outcomes Measurement Information System (PROMIS), 2018*). Four of the instruments' items capture the physical health dimension while another four assess the mental health dimension, and the final two items are for general HRQoL measurement. The principles outlined in the PROMIS Global Health Scoring Manual, which uses response pattern scoring, were followed to generate Physical Health (physical HRQoL) and Mental Health (mental HRQoL) summary scores (*Patient-Reported Outcomes Measurement Information System (PROMIS), 2021*). The PROMIS Global Health instrument has multiple response options and has an intended recall period of seven days. In order to keep recall period standardized throughout both OHIP-5 and PROMIS v.1.2 General Health questionnaires, and to gauge population health over a longer period of time, a 12-month recall period was used for this study.

# DATA ANALYSIS

## Functional relationship between OHRQoL and HRQoL

To visualize the relationships between mental HRQoL and OHRQoL and between physical HRQoL and OHRQoL, we created scatter plots and added a LOWESS (LOcally WEighted Scatterplot Smoothing) curve to investigate the functional relationship between the variables. LOWESS fits a flexible curve to the data to detect non-linear relationships. When

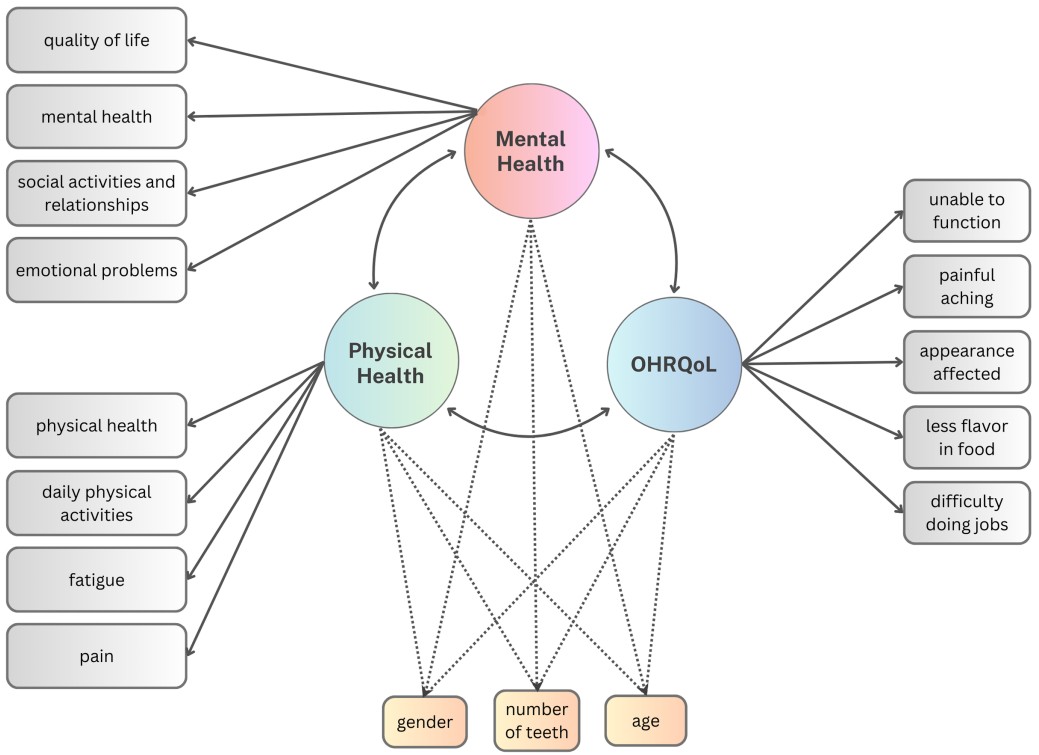

**Figure 1  Models 1 & 2 path diagram to estimate the relationship between HRQoL and OHRQoL.** The dashed paths were only included in the sensitivity model (Model 2).

a linear relationship was observed, a straight line, representing a linear relationship, was also fit to the data.

## Bivariable correlation

The linear relationship between OHRQoL and HRQoL (Physical Health and Mental Health dimensions) was summarized by calculating Spearmen and Pearson correlations among sum scores of the OHRQoL, mental HRQoL, and physical HRQoL responses.

## Structural equation models (SEMs)

A series of SEMs were generated to investigate the relationship between OHRQoL and HRQoL. We fit four models to these data by treating polytomous OHRQoL and HRQoL items as categorical, using unweighted least squares estimation. In the first two models, we separately considered the Pearson correlation between the latent variables of mental health and OHRQoL and the Pearson correlation between the latent variables of physical health and OHRQoL. The primary model (Model 1) only considered OHRQoL and HRQoL whereas the sensitivity model (Model 2) evaluated the residual correlation between OHRQoL and HRQoL after adjusting for the influence of gender, age, and teeth count (Fig. 1). Adjusting for these sociodemographic traits aligns with common practice in similar studies (*Reissmann et al., 2013*; *Baron et al., 2015*; *Barrios et al., 2015*; *Zucoloto, Maroco & Campos, 2016*; *Pakpour et al., 2016*; *Sekulić et al., 2020*).

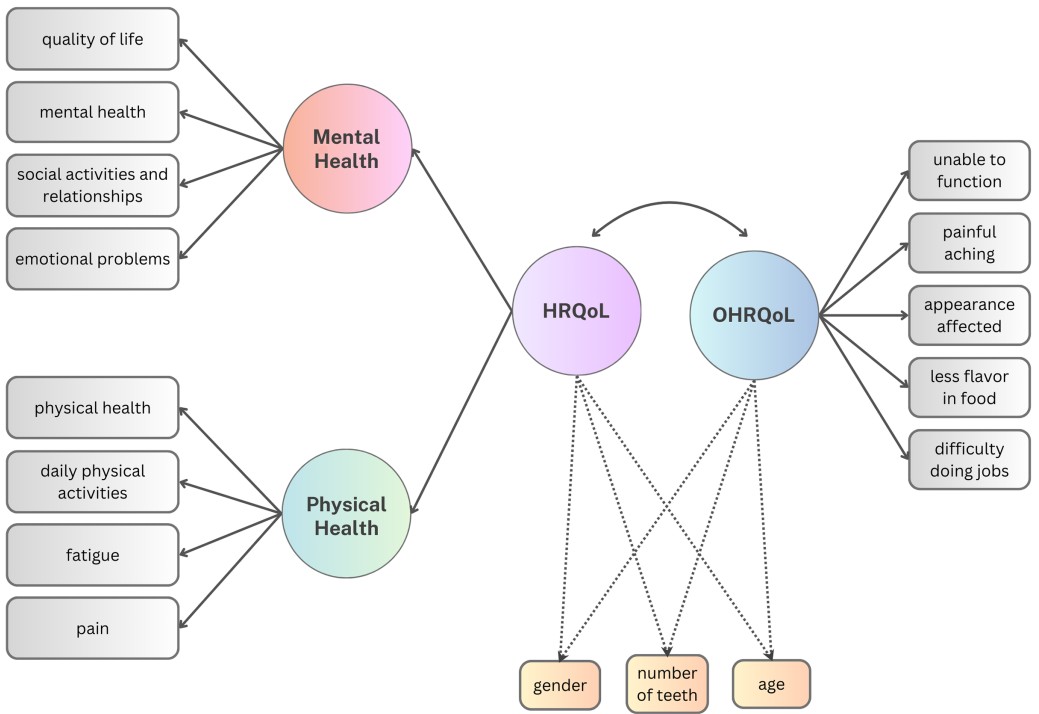

**Figure 2 Models 3 & 4 path diagram to estimate the relationship between HRQoL and OHRQoL.** The dashed paths were only included in the sensitivity model (Model 4).

In the third and the fourth model (Fig. 2), we specified a higher-order HRQoL factor and considered the Pearson correlation between the latent variables of OHRQoL and HRQoL. Again, we first evaluated only OHRQoL and HRQoL without considering additional variables (Model 3). Then, the influence of gender, age, and teeth count was investigated by adjusting the model for these covariates (Model 4).

Model fit for all four models was evaluated using the root mean square error of approximation (RMSEA), standard root mean squared residual (SRMR), comparative fit index (CFI) and Tucker-Lewis index (TLI). According to previous research, acceptable model fit guidelines include RMSEA ≤ 0.06, SRMR < 0.08 and TLI and CFI ≥ 0.95 (*Hu & Bentler, 1999*).

We used the listwise method to deal with the missing data. Data was excluded for participants who had one or more missing OHIP responses, in addition to missing any of the four responses for each of the two HRQoL domains, physical and mental health. Data loss could mostly be attributed to missing demographic information (ethnicity), rather than missing OHRQoL or HRQoL indicators. The data was initially collected using REDCap and was analyzed using Stata 17 and the lavaan package for R in later steps (*Harris et al., 2009*; *Rosseel, 2012*; *StataCorp, 2021*; *R Core Team, 2023*).

**Table 1  Participant characteristics, and instrument summary scores.**

| Participants (N = 607) | Mean (SD) or % |
|---|---|
| Age | 43.7 (17.6) |
| Sex (female) | 68.0 |
| Race | |
| White | 90.8 |
| Non-white | 9.2 |
| Ethnicity[a] | |
| Hispanic or latino | 3.0 |
| Not hispanic or latino | 89.1 |
| Education | |
| Some high school | 0.3 |
| High school graduate or GED | 7.4 |
| Some college or 2-year degree | 21.8 |
| 4-year college graduate | 31.1 |
| More than 4-year college degree | 39.4 |
| Dental insurance | |
| Employer sponsored plan | 65.6 |
| Public insurance | 5.8 |
| Self-purchased private plan | 6.1 |
| No dental insurance | 16.1 |
| Don't know/not sure/other | 6.4 |
| Number of teeth | 27.1 (3.9) |
| Removable partial and/or complete dentures | 8.0 (1.3) |
| OHIP-5 summary score[b] | 17.1 (3.3) |
| PROMIS v.1.2 global health dimension scores[c] | |
| Physical health | 16.4 (2.3) |
| Mental health | 15.7 (3.1) |

Note:
SD, standard deviation. [a]Proportion may not sum up to 100 due to missing information. GED, general educational development. OHIP, Oral Health Impact Profile. PROMIS, Patient-Reported Outcomes Information System. [b]Higher summary scores represent better OHRQoL. [c]Higher dimensions scores represent better HRQoL.

## RESULTS

### Study participants

A total of 635 were conveniently sampled. Participants were excluded if they had missing data on any of the variables of interest (OHIP-5 summary score, mental HRQoL, physical HRQoL, gender, age, and total number of teeth) for a sample size of N = 618. This data loss comprised of less than 3% of our initial sample of 635. Those who reported a gender other than male or female, or preferred not to report their gender were also excluded from the analysis due to very small numbers in these individual categories. Thus, our final sample size was N = 607. The mean age of participants was 43.7 (SD 17.6), with 68% being female (Table 1). Most participants were white (90.8%), and not Hispanic or Latino (89.1%). The average OHIP = 5 summary score of 17.1 (SD 3.3), out of a possible 20, indicating that this population did not suffer substantially from oral health impacts as a higher score

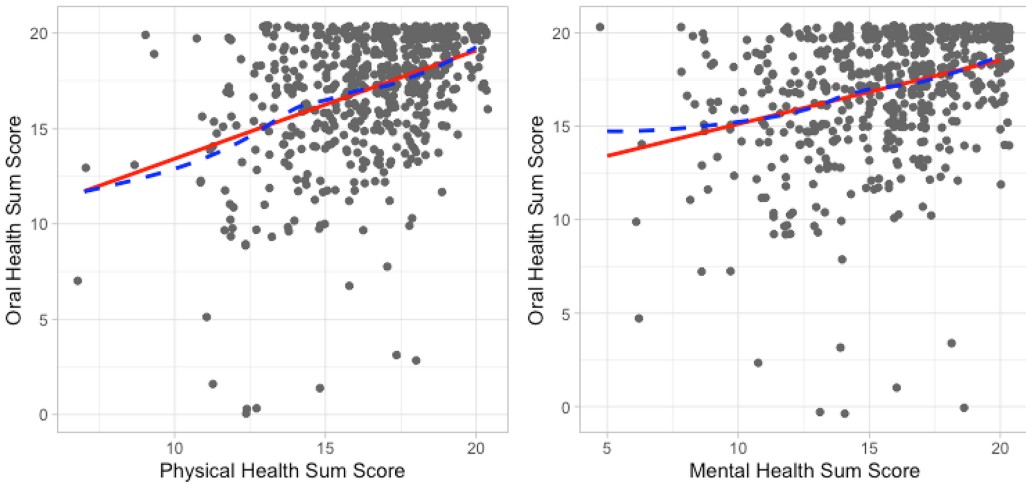

**Figure 3 Functional relationship between mental health and OHIP sum scores as well as between physical health and OHIP sum scores characterized by a flexible dashed line and a solid straight line.**

indicates better OHRQoL. Physical and mental health dimensions scores had means of 16.2 (SD 2.3) and 15.8 (SD 3.1), respectively.

## Functional relationship between HRQoL and OHRQoL

The best flexible fit and the best linear fit for the relationships between mental HRQoL and OHRQoL, as well as between physical HRQoL and OHRQoL scores, were very close. Figure 3 indicates that these variables had a linear functional form, allowing the use of SEM to correlate the latent variables and the Pearson correlation coefficient to correlate summary scores.

## Correlation between HRQoL and OHRQoL

### Bivariable correlation

An assessment of the relationship between physical HRQoL and OHRQoL yielded a Pearson correlation of r = 0.39 (95% CI [0.32–0.46]) and a Spearman correlation of 0.36 (95% CI [0.29–0.42]). For the mental HRQoL and OHRQoL relationship, a slightly lower Pearson correlation of r = 0.32 (95% CI [0.25–0.39]) and a Spearman correlation of 0.34 (95% CI [0.26–0.41]) was observed. The similarity between the Pearson and the Spearman correlations, taken together with the high similarity between linear and flexible lines in Fig. 3, suggests that the relationships between these summary scores are well-characterized by linear functions.

### Structural equation models

Like the bivariable correlations, the SEM models also showed that OHRQoL correlated slightly higher with physical HRQoL than with mental HRQoL, r = 0.55 and r = 0.43, respectively (Table 2). As expected, the OHRQoL correlation with the higher order HRQoL factor was in between the two results (r = 0.51/0.52). The 95% confidence interval widths of 0.09 and smaller indicated sufficient precision around point estimates. The size

**Table 2 Summary of SEM models' results correlating OHRQoL with HRQoL.**

| Model | Adjusted for co-variates? | Variable correlated with OHRQoL | Estimated correlation [95% CI] | (Partial) $R^2$ effect size | RMSEA | SRMR | CFI | TLI |
|---|---|---|---|---|---|---|---|---|
| 1 | No | Physical HRQoL | 0.55 [0.50–0.59] | 0.30 | 0.057 | 0.058 | 0.99 | 0.98 |
| | | Mental HRQoL | 0.43 [0.40–0.46] | 0.18 | | | | |
| 2 | Yes | Physical HRQoL | 0.53 [0.49–0.57] | 0.28 | 0.047 | 0.056 | 0.99 | 0.99 |
| | | Mental HRQoL | 0.43 [0.40–0.46] | 0.18 | | | | |
| 3 | No | HRQoL | 0.52 [0.50–0.55] | 0.27 | 0.057 | 0.058 | 0.99 | 0.98 |
| 4 | Yes | HRQoL | 0.51 [0.48–0.54] | 0.26 | 0.046 | 0.056 | 0.99 | 0.99 |

**Note:**
Effect size is the square of the estimated Pearson correlation. When the model adjusts for covariates, this is a partial $R^2$. RMSEA, root mean squared error of approximation. SRMR, standardized root mean squared residual. CFI, comparative fit index. TLI, Tucker-Lewis index. OHRQoL, oral health-related quality of life. HRQoL, health-related quality of life.

of these correlations was "medium" to "large" as per Cohen's effect size guidelines (*Cohen, 1988*, *1992*). Adjustment for gender, age, and the number of teeth did not notably change the observed correlation estimates. All four models fit well, meeting guideline values.

## DISCUSSION

This cross-sectional study showed the magnitude of the association between OHRQoL and HRQoL to be r = 0.52 (95% CI [0.50–0.55]), indicating that the two constructs shared 27% of their information. According to guidelines for the interpretation of a correlation's magnitude, this study provides evidence of a "large" correlation between OHRQoL and HRQoL in a USA-based general adult population (*Cohen, 1988*, *1992*). Our study results are the first to indicate the relationship between the two constructs over longer periods of time.

Although the populations and instruments used varied, we made efforts to compare our results to the existing literature in a standardized manner to further understand and contextualize the magnitude of the correlation between OHRQoL and HRQoL (Table 3; *Persson et al., 2009*; *Zimmer et al., 2010*; *Inoue et al., 2011*; *Östberg & Hall-Lord, 2011*; *Reissmann et al., 2013*; *Wickert et al., 2014*; *Baron et al., 2015*; *Barrios et al., 2015*; *Zucoloto, Maroco & Campos, 2016*; *Pakpour et al., 2016*; *Kuo et al., 2018*; *Sekulić et al., 2020*; *Rojas-Alcayaga et al., 2022*; *Purisinsith et al., 2022*). The reported correlations among studies sometimes have a negative sign. While higher HRQoL scores, such as the SF-36 and the SF-12, indicate better HRQoL, higher OHRQoL scores, such as the OHIP-based score, represent worse OHRQoL. In the present study, we recoded the OHIP-score so that a higher score indicated better OHRQoL. Hence, when we compare literature findings with our results, we interpret the absolute magnitude of the correlation.

Bivariable correlations, such as Pearson and Spearman rank correlations are attenuated compared to SEM derived correlations. The attenuation is not small. We observed correlation differences of 0.11 and 0.16. Comparing bivariable correlations with SEM-derived correlations showed a similar magnitude (*Reissmann et al., 2013*).

The absolute magnitude of reported bivariable correlations were similar to our bivariable correlations, regardless of population or recall period pointing to a similarity if

**Table 3 Studies measuring correlation between OHRQoL and HRQoL.**

| Title | First author (year published) | OHIP version | HRQoL instrument | Sample size | Population | Recall period | Method | Adjustment performed | OHRQoL/ GHRQoL | OHRQoL/ HRQoL (physical) | OHRQoL/ HRQoL (mental) |
|---|---|---|---|---|---|---|---|---|---|---|---|
| Association between oral health-related and health- related quality of life. | Sekulic et al. (2020) | OHIP-49 | PROMIS v.1.1 global health | 2,076 | Dental patients (USA) | 1 Month | SEM (without HRQoL factor) | Model 3: age, gender, and depression | | 0.52 | 0.47 |
| | | | | | | | SEM (with HRQoL factor) | Model 2: none | 0.56 | 0.55 | 0.51 |
| | | | | | | | SEM (without a HRQoL factor) | Model 1: none | | 0.55 | 0.51 |
| Illness experience and quality of life in sjogren syndrome patients. | Rojas-Alcayaga et al. (2022) | OHIP-14Sp (Spanish) | EuroQoL-VAS | 31 | Women with Sjogren's syndrome patients (Chile) | | Bivariate | | -0.343 | | |
| Assessing a conceptual model with both oral health and health related quality of life in community-dwelling elders. | Kuo et al. (2018) | OHIP-14T (Taiwanese) | WHOQoL-Bref | 517 | Elderly general population (Taiwan) | 12 Months | Bivariate | | -0.39 | | |
| Association between general and oral health-related quality of life in patients treated for oral cancer. | Barrios et al. (2015) | OHIP-14 | SF-12 | 142 | Oral cancer patients (Granada) | 1 Month | Bivariate | Age and sex | | 0.41 | 0.49 |
| The Canadian systemic sclerosis oral health study II: the relationship between oral and global health-related quality of life in systemic sclerosis. | Baron et al. (2015) | OHIP-49 | SF-36 | 156 | Scleroderma and systemic sclerosis (SSc) patients (Canada) | | Bivariate | | | -0.29 | -0.30 |
| | | | | | | | Bivariate | Age, sex, race, education, smoking, disease duration, and disease severity | | 0.24 | 0.31 |
| Sensitivity to change of oral and general health-related quality of life during prosthodontic treatment. | Wickert et al. (2014) | OHIP-G (German) | SF-36 | 166 | Dental patients in a prosthodontic clinic (Germany) | 1 Month | Bivariate | | | -0.32 | -0.37 |
| Association between perceived oral and general health. | Reissmann et al. (2013) | OHIP-49 | SF-36 | 311 | Dental patients (Germany) | 1 Month | Bivariate | | | | |
| | | | | | | | SEM | No | 0.4 | 0.31 | 0.40 |
| | | | | | | | SEM | Age, gender and level of depression | 0.24 | | |
| | | | | 811 | General population (Germany) | | Bivariate | | | 0.45 | 0.45 |
| | | | | | | | SEM | No | 0.54 | | |
| | | | | | | | SEM | Age, gender and level of depression | 0.28 | | |

(Continued)

| Title | First author (year published) | OHIP version | HRQoL instrument | Sample size | Population | Recall period | Method | Adjustment performed | Correlation | | |
|---|---|---|---|---|---|---|---|---|---|---|---|
| | | | | | | | | | OHRQoL/ GHRQoL | OHRQoL/ HRQoL (physical) | OHRQoL/ HRQoL (mental) |
| Denture quality has a minimal effect on health-related quality of life in patients with removable dentures. | Inoue et al. (2011) | OHIP-J (Japanese) | SF-36 | 171 | Dental patients with removable dentures (Japan) | 1 Month | Bivariable linear regression | Denture stability | | −0.91 | −1.24 |
| | | | | | | | Bivariable linear regression | Denture aesthetics | | −0.84 | −1.31 |
| Oral health-related quality of life, a proxy of poor outcomes in patients on peritoneal dialysis. | Purisinsith et al. (2022) | OHIP-14 | SF-12 | 222 | Peritoneal dialysis patients (Thailand) | "A few months" | Bivariate (Spearman's correlation coefficient) | | | −1.69 | −0.18 |
| Association between oral health-related and general health-related quality of life in subjects attending dental offices in Germany. | Zimmer et al. (2010) | OHIP-14 (German) | SF-12 | 12,392 | Dental patients (Germany) | | Bivariate (Pearson's correlation coefficient) | | | 0.309 | 0.318 |
| Oral health-related quality of life in Iranian patients with spinal cord injury: a case-control study. | Pakpour et al. (2016) | OHIP-14 (Iranian) | SF-36 | 203 | Spinal cord injury patients (Iran) | | Hierarchical liner regression | Age, gender, sex, education, clinical variables, oral care behavior variables, anxiety and depression | | −0.275 | −0.191 |
| Oral health-related quality of life in older Swedish people with pain problems. | Östberg & Hall-Lord (2011) | OHIP-14 SC (simple count sum score method) | SF-12 | 186 | Elderly general population with pain problems (Sweden) | 12 Months | Bivariate (Spearman's correlation coefficient) | | | −0.091 | −0.375 |
| | | OHIP-14 ADD (additive sum score method) | | | | | | | | −0.084 | −0.41 |
| Association of perceived quality of life and oral health among psychiatric outpatients. | Persson et al. (2009) | OHIP-14 (Swedish version) | SF-12 | 113 | Patients with severe mental illness (Sweden) | | Bivariate (Spearman's correlation coefficient) | | | −0.30 | −0.20 |
| Impact of oral health on health-related quality of life: a cross-sectional study. | Zucoloto, Maroco & Campos (2016) | OHIP-14 (Portuguese version) | SF-36 | 1,007 | Dental population (Brazil) | | SEM | Age, pain and presence of chronic disease | −0.53 | | |

SEM would have been used (*Zimmer et al., 2010*; *Wickert et al., 2014*; *Kuo et al., 2018*; *Rojas-Alcayaga et al., 2022*). Therefore, it can be expected that the deattenuated correlation would reach a similar magnitude as reported in this study and in the only three other studies that used SEM (*Reissmann et al., 2013*; *Zucoloto, Maroco & Campos, 2016*; *Sekulić et al., 2020*).

Adjustment for other factors, such as depression, should lower the HRQoL-OHRQoL correlation; however, results were mixed. While the present study found a minimal decrease when depression was accounted for, the only other study that adjusted an SEM—derived correlation showed a substantial decrease that was similar for dental patients and general population subjects.

The influence of instrument length was difficult to investigate. A study using OHIP-49 instead of OHIP-5 showed similar results. However, OHIP-5 and OHIP-49 summary scores correlated with r = 0.93, indicating that, at least on the summary score level, OHIP-5 measured OHRQoL well (*John, 2022*; *John et al., 2022*). For SF-12 and SF-36, correlations are also very large (*Müller-Nordhorn, Roll & Willich, 2004*).

For the investigation of influence of the recall period, available data was limited. Many studies did not report the recall period at all. However, our study's finding using the longest recall period (1 year) showed very similar findings with studies using a 1-month recall period (*Reissmann et al., 2013*; *Sekulić et al., 2020*). This may be due to the fact that the recall period for OHIP does not have a very large influence. Previous studies have shown that a 1-month recall period is similar to a 12-month period, and a 1-week recall period is similar to a 1-month period (*Waller et al., 2016*).

Overall, these results indicate that the OHRQoL-HRQoL relationship is "medium" to "large" according to Cohen's guidelines for magnitude interpretation (*Cohen, 1988*, *1992*). That the correlation is small, is unlikely. A meta-analysis based on a systematic review is necessary to summarize the evidence and to provide more insight about factors influencing the correlation, for example, the homogeneity of the correlation across cultures.

## STRENGTHS AND LIMITATIONS

This study used a practical method to measure OHRQoL and HRQoL with widely used instruments. PROMIS' Global Health instrument comes from a family of measures that have greater precision than most conventional measures. Many PROMIS instruments, such the Global Health instrument, have short forms and computer-adaptive testing forms, increasing the applicability of such instruments. OHIP-5 is the shortest instrument of a family of instruments with different lengths that are widely used (*Yu et al., 2023*). These instruments are globally available, *Ingleshwar & John (2023)* from low-resource setting (*Lawal & Omara, 2023*) to randomized trials, *Reuter-Selbach, Su & Faggion (2023)*, *Tao et al. (2023)* and used by different oral health care providers (*Chanthavisouk et al., 2022*) in adults (*Mittal et al., 2019*; *Rener-Sitar et al., 2021*) and children (*Shayestehpour et al., 2022*).

Our findings are precise. With fewer OHIP variables being present in the OHIP-5, attenuation of the correlation, and greater error, was expected in comparison to the studies that used the OHIP-49 for OHRQoL measurement. This was not necessarily the case, as

our study found a similar, or slightly greater correlation between the constructs when compared to those that used the more comprehensive OHIP-49 (*Reissmann et al., 2013*; *Sekulić et al., 2020*).

Our results fit well with literature findings even if the constructs HRQoL and OHRQoL can both be captured with a number of methodological options which include a variety of available instruments, version, recall periods to select from, and analytic approaches.

It is a strength of our study that this is the first study to utilize OHIP-5 to assess the association with longer recall periods that are often used to assess public oral health impact which is in contrast to previous studies that utilized longer versions of the OHIP (*Zimmer et al., 2010*; *Reissmann et al., 2013*; *Sekulić et al., 2020*). Nevertheless, per *Cohen*'s *(1988, 1992)*, the magnitude of the correlation remains positive between the constructs, regardless of the OHIP version used.

It is important to acknowledge a limitation related to the voluntary nature of participation in our study. It is possible that individuals with better oral and/or general health were more inclined to participate. This self-selection bias may introduce a potential source of selection bias in our results, as those who chose to take part may not be entirely representative of the broader population.

## Relevance

OHRQoL refers to the impact that one or more oral health conditions may have on a person's well-being, while HRQoL encompasses a broader range of physical and mental health dimensions. This is due to the various ways in which oral health can impact physical and mental well-being, such as orofacial pain, oral function, orofacial appearance, and broader psychosocial effects, *i.e.*, the Dimensions of OHRQoL. These dimensions provide a comprehensive view of the impact of oral health on a person's overall quality of life and help in determining the appropriate interventions and treatments needed to improve oral health.

While our study contributes to the evidence for the relationship between oral and overall health, the need for a systematic review with meta-analysis became clear for a more in-depth assessment of the relationship by harvesting all previous findings reproducibly and comprehensively to summarize them analytically. This study consolidated the OHRQoL and HRQoL association and found a "large" correlation between the two constructs despite the use of an OHRQoL measurement tool that includes only five-items. This indicates a substantial link between oral and overall health exists, and that the size of this relationship may be large enough for practical application for measuring these constructs in a variety of settings.

Dental professionals can use insights gained through practical OHRQoL measurement with the OHIP-5 to inform the broader construct, HRQoL, of their patients. Such insights on the correlation between the constructs could also provide a practical pathway for OHRQoL and HRQoL measurement in a variety of settings. Additionally, healthcare providers should consider the relationship between oral health and overall health, assessing a patient's oral health as part of their overall health assessment, and vice versa. By taking a holistic approach to healthcare and recognizing the connection between oral

and overall health, providers can help their patients maintain good health and well-being for a lifetime.

## CONCLUSIONS

Utilizing the briefest version of the OHIP questionnaire, OHIP-5, and a 12-month recall period, the association between the constructs OHRQoL and HRQoL was substantial. We found that the relationship between the two constructs goes beyond a minor, or "small" correlation. Future research, such as a meta-analysis, that collectively examines the results of the existing literature could help provide a much-needed, pooled estimate of the magnitude of the association between OHRQoL and HRQoL, two crucial aspects of patient's well-being.

## ACKNOWLEDGEMENTS

We would like to thank the following people for assistance with recruitment, and data collection at the State and County fairs–Drew Christianson, Melissa Durbin, Keeley Flavin, Pelisherto Her, Jodi Pouliot, Dr. Yvette Reibel, Dr. Karl Self, Danae Seyffer, Dr. Cynthia Stull, Rachel Thelen, Mercedes VanDeWiele, and Eugene Vang. We are particularly grateful to Dr. Priscilla Flynn, the Principal Investigator of the University of Minnesota, School of Dentistry's "Tooth or Consequences" 2022 State Fair project that provided the data for the current study. Additionally, we would like to extend our gratitude to Nicole Theis-Mahon at University of Minnesota, Health Sciences Library, for her invaluable expertise in navigating the resources and databases which was pivotal to the success of this study.

### Funding

This research was supported by the National Institutes of Health's National Center for Advancing Translational Sciences, grant UL1TR002494. The funders had no role in study design, data collection and analysis, decision to publish, or preparation of the manuscript.

### Grant Disclosures

The following grant information was disclosed by the authors:
National Institutes of Health's National Center for Advancing Translational Sciences: UL1TR002494.

### Competing Interests

The authors declare that they have no competing interests.

### Author Contributions

- Danna R. Paulson conceived and designed the experiments, performed the experiments, analyzed the data, prepared figures and/or tables, authored or reviewed drafts of the article, and approved the final draft.

- Phonsuda Chanthavisouk conceived and designed the experiments, performed the experiments, analyzed the data, prepared figures and/or tables, authored or reviewed drafts of the article, and approved the final draft.
- Mike T. John conceived and designed the experiments, performed the experiments, analyzed the data, prepared figures and/or tables, authored or reviewed drafts of the article, and approved the final draft.
- Leah Feuerstahler conceived and designed the experiments, performed the experiments, analyzed the data, prepared figures and/or tables, authored or reviewed drafts of the article, and approved the final draft.
- Xing Chen conceived and designed the experiments, performed the experiments, analyzed the data, prepared figures and/or tables, authored or reviewed drafts of the article, and approved the final draft.
- Aparna Ingleshwar conceived and designed the experiments, performed the experiments, analyzed the data, prepared figures and/or tables, authored or reviewed drafts of the article, and approved the final draft.

## Human Ethics

The following information was supplied relating to ethical approvals (*i.e.*, approving body and any reference numbers):

University of Minnesota IRB granted approval for STUDY00016028.

## Data Availability

The survey results are available in the Supplemental File.

## Supplemental Information

Supplemental information for this article can be found online at http://dx.doi.org/10.7717/peerj.17440#supplemental-information.

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
