# Peer review of "Linking patient-reported oral and general health-related quality of life"

_PeerJ, doi:10.7717/peerj.17440_

## Round 0.1 · original submission · Minor Revisions

Authors should review the article according to the reviewers' criticisms and submit a new version

·

Basic reporting

Thank you for inviting me to review the article titled “Linking patient-reported oral and general health-related quality of life”. In general, the article is well written. However, some of the concerns/corrections need to be done.
Title:
1. Kindly include the study design type.
Abstract:
2. Authors can specify the statistical test used to find the link. Since the abstract is well within the word count of peerj, this may be considered.
Introduction:
3. It’s fairly written. However, some of the references that are more than five years old can be reduced.
4. Authors can emphasize the relationship between oral health and general health rather than description of the tools.
5. Except for 12-month recall strategies, is there any other specific rationale for the study?
Methods:
6. Kindly mention more details about the inclusion and exclusion criteria
7. I suggest looking into STROBE/CROSS statements to improve the methodology presentation.
8. The authors included visually impaired adults also. How would it impact the overall results (bias/confounding, etc.)? As the visually impaired patients have poor oral OHRQoL
Results:
9. Well-presented
Discussion:
10. Again, the same suggestions. Kindly update references (OHRQoL and general QoL have changed dramatically over the period of time) and modify the discussion accordingly.
11. If authors wish to maintain the same references, kindly add the discussion about the changes/trends in the link/association.

Experimental design

The authors used the proper methods to evaluate the intended objectives. However, it required more explanation.

Validity of the findings

1. The validity of findings clearly benefits/adds to the existing literature
2. The provided data is robust and statistically sound
3. Discussion parts need improvement (as per the above-mentioned comments)

Reviewer 2 ·

Basic reporting

Though on the whole, the manuscript was well-written, introduction section may be modified to be more concise.
Consider a more engaging way to introduce the topic, perhaps highlighting the impact of poor oral health on daily life.
From line 55 to 60, the authors should first give more illustrations for the meaning of“r=0.24 and r=0.28”to facilitate the reader understanding these index.
From line 72 to 87, please briefly summarize the key findings and consider combining some examples. For instance, it could shorten as:
"...Disease-specific research also highlights the connection between oral health and overall well-being. Studies involving temporomandibular disorder (TMD) patients, head and neck cancer patients, and individuals with dentofacial deformities all demonstrate positive correlations between OHRQoL and HRQoL. "
Consider adding a sentence at the end that briefly mentions the potential impact of the study. For example:
"By investigating the OHRQoL-HRQoL relationship over a longer period, this study aims to provide valuable insights for healthcare professionals and contribute to the development of more comprehensive healthcare strategies."

Experimental design

Please briefly explain the rationale for using a 12-month recall period for both instruments despite the PROMIS instrument having a recommended 7-day recall.

Validity of the findings

In the discussion, please explain more on the limitations of instruments and how it might affect the results and the mixed findings on the effect of adjusting for confounding variables and the need for further investigation.

Additional comments

The illustration in figure1 and 2 should be more clear, such as” gen_daily””ohip_less” etc, it may be better to illustrate these in the figure rather than in figure legends.

Reviewer 3 ·

Basic reporting

no comment

Experimental design

- how accurate is the physical oral health measurement? Is it the standard procedure to have one dentist perform the test? What was the inter-rater agreement from the previous studies? You could provide one or two sentences to describe that.
- both OHRQoL and HRQoL are discrete variables. To analyze a bivariate correlation, I think the Spearman correlation test would be more suitable than the Pearson correlation test.
- It’s unclear why gender, age, and teeth count were adjusted in the SEM. Could you include some background to indicate that these factors are associated with either OHRQoL or HRQoL?

Validity of the findings

- Apart from gender, age and teeth count, are there any other confounders that you should adjust for but do not have relevant data? You could add one or two sentences in the limitation section.

Additional comments

no comment

---

## Round 0.2 · accepted · Accept

The manuscript is ready for publication

Reviewer 2 ·

Basic reporting

no comment

Experimental design

no comment

Validity of the findings

no comment

Additional comments

no comment

Reviewer 3 ·

Basic reporting

The authors have sufficiently addressed my comments.

Experimental design

The authors have sufficiently addressed my comments.

Validity of the findings

The authors have sufficiently addressed my comments.